# Acute Myocardial Infarction among Young Adult Men in a Region with Warm Climate: Clinical Characteristics and Seasonal Distribution

**DOI:** 10.3390/ijerph17176140

**Published:** 2020-08-24

**Authors:** Chiao-Yu Shih, Min-Liang Chu, Tsung-Cheng Hsieh, Han-Lin Chen, Chih-Wei Lee

**Affiliations:** 1Department of Physical Therapy, Tzu Chi University, Hualien 97004, Taiwan; cysjoey@gms.tcu.edu.tw; 2Institute of Medical Sciences, Tzu Chi University, Hualien 97004, Taiwan; stin547598@gmail.com (M.-L.C.); tchsieh@mail.tcu.edu.tw (T.-C.H.); 3Center for General Education, Tzu Chi University of Science and Technology, Hualien 97004, Taiwan; hanlin@tcust.edu.tw

**Keywords:** acute myocardial infarction, young adults, clinical characteristics, seasonal distribution

## Abstract

The aim of this cross sectional study was to investigate the influence of the seasons on acute myocardial infarction (AMI) among young adult among young adults aged <45 years compared to old adults aged ≥45 years. The seasonal distribution of AMI hospital admissions among young adult men in eastern Taiwan was assessed. Data were extracted from 1413 male AMI patients from January 1994 to December 2015, including onset date, the average temperature (Tave) on the date of AMI hospitalization (AMI-Tave), and conventional risk factors, notably smoking, diabetes, hypertension, total cholesterol, high-density lipoprotein cholesterol, low-density lipoprotein cholesterol, triglycerides, and body mass index (BMI). The 1413 cases were divided into two groups: the young group (*n* = 138, <45 y/o) and the older group (*n* = 1275, ≥45 y/o). The differences between groups were examined. Logistic regression analyses were used to evaluate the associations between the seasons and the AMI hospitalization among the young group. The young group showed significantly higher percentage of smokers, BMI, total cholesterol levels, and triglycerides levels but lower percentage of diabetes and hypertension than the older group (*p* < 0.05). AMI hospitalization in winter was significantly greater compared to the other seasons among the young group (*p* < 0.05). Winter hospitalization was significantly associated with the young group relative to the older group (adjusted OR 1.750; 95% CI 1.151 to 2.259), while winter AMI-Tave in the young group was similar to that in the older group. Young adult men diagnosed with AMI are more likely than older adult men to be smokers, obese, and show an onset dependent on winter but not low-temperature in a region with a warm climate.

## 1. Introduction

Emerging evidence shows that incidences of cardiovascular disease have been either steady or increasing among young adults aged <45 years over the past few decades in contrast to the downward trend in older adults [1]. Acute myocardial infarction (AMI), a severe manifestation of coronary heart disease, occurs less frequently in young adults, but it can cause devastating consequences for young patients and their family [2]. Evidence indicates that young adults diagnosed with AMI are predominately men who are smoker, obese, consume a poor diet, physically inactive, and are often unaware of warning signs and high-risk situations before symptom onset [1,2,3,4]. Previous studies revealed an increase in AMI among men, markedly younger men, in recent years [5,6]. Prevention measures for AMI may need to give great attention on the identification of potential risks for AMI in adult men, especially young adult men.

The susceptibility toward AMI is influenced by the combined effects of long-term abnormal regulation in the cardiovascular system induced by the interaction of genetic defects, unhealthy lifestyle choices, environmental determinants and short-term modification in response to contextual triggers [7,8]. Seasonal changes are periodically short-term events and have potential for modulation of susceptibility to AMI. This is supported by the findings that seasonal patterns show in the incidence of cardiovascular diseases, conventional risk factors, and behavioral risk factors, such as fat-rich dietary intake and physical inactivity [9,10,11,12].

Environmental and personal factors have an influence on the seasonality of AMI. Most epidemiological studies reported a winter peak and summer trough in the incidence of AMI [9,13]. However, in regions with warm climate, some studies found that AMI hospitalizations showed a summer peak or no seasonality [14,15]. Effects of age and co-morbid risk factors also interact with the seasonality of AMI in different climatic regions. Several studies reported inconsistent seasonal patterns of AMI hospitalizations between the elderly and the non-elderly and diabetes mellitus (DM) patients and non-DM patients [14,16]. A previous study from western Taiwan indicated that advanced age tended to influence season-related incidences of AMI in the relatively cool region, while comorbid metabolic dysfunction had an influence in the relatively warm region [17]. There was lack of data regarding seasonality of AMI among young adults <45 years within the literature, which is the area of study that this paper sought to address.

The purpose of this study was to investigate the influence of the seasons on AMI hospital admissions among young (<45 years) compared to older (≥45 years) adult men in eastern Taiwan, a region situated in the tropics and subtropics. Differences in clinical characteristics and association of season-related occurrences of AMI were also examined between young and older adult men.

## 2. Materials and Methods

### 2.1. Study Cases

This was a retrospective and cross-sectional study with study cases being obtained from medical records. Hualien Tzu Chi Hospital is a medical center that accepts most of the transferred AMI patients in eastern Taiwan. From January 1994 to December 2015, data were accessed on 3378 individuals admitted to the hospital with a first-time diagnosis of AMI, including 2287 men (191 <45 y/o and 2096 ≥45 y/o) and 1091 women (27 <45 y/o and 1064 ≥45 y/o). The female population in the <45 y/o age group was too small for meaningful statistical analysis over four seasons. As such, this study did not include female cases. Of the 2287 men, 1413 with complete data were included in the analysis, but 875 with incomplete records, whose date of admission could not be accurately assessed and lacked one or more records of conventional risk factors were excluded. Comparison of variable distribution between the included cases and the excluded cases is shown in Appendix A. The 1413 included cases were assigned to two groups: the young group (*n* = 138, <45 y/o) and the older group (*n* = 1275, ≥45 y/o). AMI was diagnosed by cardiovascular specialists based on characteristic clinical history, serial changes on the electrocardiogram (ECG), and an increase in cardiac enzymes.

This study was approved by the Research Ethics Committee at the Buddhist Tzu Chi General Hospital (IRB097-98).

### 2.2. Data Collection of Clinical Characteristics

The patients’ demographic information, diagnosis, the date of AMI symptom onset, coronary angiography (CAG), and conventional risk factors were extracted from medical records. The data regarding conventional risk factors were expressed as the prevalence of current cigarette smoking, diabetes mellitus (DM), hypertension, the values of total cholesterol (TC), triglycerides (TG), high-density lipoprotein cholesterol (HDL-C), low-density lipoprotein cholesterol (LDL-C), and body mass index (BMI). The date of AMI symptom onset was determined by cardiovascular specialists based on the initiation of ischemic-related symptoms leading to evolved AMI. Coronary artery stenosis was defined as a ≥50% reduction in the internal diameter of coronary arteries. Diabetes was defined as fasting blood sugar ≥ 140 mg/dL (before 1997) or 126 mg/dL (after 1997) or the use of specific treatment. Hypertension was defined as systemic blood pressure ≥ 140/90 mmHg or a history of previous treatment.

### 2.3. Weather Data and Season

For the purpose of this study, the four seasons were defined as follows: spring = 1 March to 31 May, summer = 1 June to 31 August, autumn = 1 September to 30 November, and winter = 1 December to 28–29 February. The average ambient temperature in each season (Season-Tave) from January 1994 to December 2015, and on the date of AMI hospitalization (AMI-Tave) was obtained from the Taiwan Central Weather Bureau. The AMI hospitalization rate and average temperature at date of onset were assessed according to the seasons.

### 2.4. Statistical Analysis

Continuous variables were evaluated the skewness using the Kolmogorov–Smirnoff test. The data were expressed as mean ± SD and evaluated the group difference by the student’s test if the data were not skewed. For the skewed cases, the Mann–Whitney test was performed to evaluate the difference between groups, and expressed as median and interquartile range (IQR). For the categorical variables, the chi-square test was used to evaluate the group differences and expressed as frequency with proportion. The multivariate logistic regression analysis was performed to assess the influence of the seasons on the age (the young/the older) of study cases at AMI hospital admissions. Age group (the young/the older) was set as the dependent variable in the regression models. Four logistic regression models were built for assessing the seasonal effects on the age of onset of AMI by using winter/non-winter, spring/non-Spring, summer/non-summer, and fall/non-fall as the main predictor variables, respectively. Clinical characteristic factors which reached statistical differences between the young and older groups were also included as predictor variables in the model for adjustment of their effects. All *p* values were two-sided with a value < 0.05 considered statistically significant. Statistical analysis was performed with SPSS for Windows version 17.0 (IBM, Armonk, NY, USA).

## 3. Results

### 3.1. Comparison of Clinical Characteristics between the Young and the Older Groups

Table 1 showed the demographics, angiographic findings, and conventional risk factors in the young group and the older group. The young group included 138 cases with mean age at 39.1 ± 5.2 years, accounting for 9.7% of total cases, and the older group included 1275 cases with mean age at 66.7 ± 11.1 years. The young group showed reduced incidence of stenosed coronary arteries compared to the older group (*p* = 0.001). In the young group, 60.2% had less than two stenosed coronary arteries, but 53.3% of the older group had more than two. Smoking rate in the young group (71.1%) was greater than that in the older group (51.1%) (*p* < 0.001). The prevalence of DM in the young group (20.3%) was lower than that in the older group (36.8%). The prevalence of hypertension in the young group (37.7%) was lower than that in the older group (66.0%). The levels of TG, TC, and BMI in the young group (140.0 ± 139.5 mg/dL, 183.0 ± 62.0 mg/dL, and 26.8 ± 5.4 kg/m^2^) were higher than those in the older group (106.0 ± 91.0 mg/dL, 172.0 ± 55.0 mg/dL, and 24.7 ± 4.8 kg/m^2^). The young group showed levels of HDL-C down to 38.0 ± 17.3 mg/dL and levels of LDL-C up to 107.5 ± 58.4 mg/dL.

### 3.2. Comparison of AMI Hospitalizations between the Seasonal Distribution

Winter showed significantly higher AMI hospitalizations compared to spring, summer, and fall in the young group (*p* < *0*.05). Winter AMI-Tave was 18.1 ± 3.2 °C in the young group and 17.9 ± 2.9 °C in the older group (Figure 1). Winter-Tave was 18.1 ± 2.2 °C.

### 3.3. Association of the Seasons with AMI Hospitalization among the Age Groups

Four logistical regression models were performed to assess the adjusted association of the season (winter/non-winter, spring/non-spring, summer/non-summer, and fall/non-fall, respectively) with the age group (the young/the older) (Table 2). The predictor variables of adjustment were clinical characteristic factors which reached statistical difference between two age groups, including smoking, DM, hypertension, TC, TG, BMI, and the number of stenosed vessels. After adjustment for the clinical characteristic factors, winter showed significantly and independently associated with the AMI hospitalizations in the young group (adjusted odd ratio (OR) 1.750; 95% CI 1.151 to 2.259).

## 4. Discussion

The major findings of this study were that male AMI patients under the age of 45 years had different characteristics in the severity of abnormal coronary arteries and risk factors, including smoking, BMI, DM, hypertension, TG, and TC, from those over the age of 45 years. Winter-linked precipitation of AMI was highly associated with the young adult men (<45 years) when compared to the older adult men (≥45 years), while the average temperature on the date of onset during winter was similar in both groups.

AMI occurred less frequently in young adults aged <45 years. Consistently, the young group accounted for less than 10% of all cases in this present study. A case–control study on smokers in 52 countries revealed the young adult men had higher population attributable risks for AMI compared to the middle-aged or older adult men [18]. Previous studies also revealed that cigarette smokers had large waist circumference, relatively high TG, and low HDL-C [19,20]. Accordingly, the young group of AMI patients in this study had a high prevalence of smoking, a high level of BMI and TG, and a low average level of HDL-C below 40 mg/dL. According to the US National Cholesterol Education Program Adult Treatment Panel III, HDL-C less than 40 mg/dL represents a high-risk for men to develop coronary heart disease [21]. Moreover, evidence supports that a higher BMI is a hazard for developing coronary heart diseases in the young population [22]. As shown in previous and this present study, the young group of AMI patients had reduced severity of diseased coronary arteries and comorbidity of hypertension and diabetes. Young adult men may suffer AMI due to some form of coronary obstruction and not necessarily due to a severe atherosclerotic condition.

Seasonal variation in the incidence of AMI has been noted across a wide range of populations and climate zones. The seasonal pattern displaying a peak in winter and a trough in summer was consistently reported from epidemiological studies concerning AMI in the regions with cold and temperate climate. However, seasonality of AMI was inconsistent in previous reports from warm climate regions. A relatively high hospital admission of AMI during summer was documented in Dallas, Texas, a region with mild winters and very hot summers [12]. In Taiwan, an island with subtropical and tropical climate zones, an absence of seasonal variation in hospital admission of AMI was claimed by a previous study [13]. The previous population-based study from western Taiwan found that hospital admissions of AMI in the cold season were significantly greater than those in the hot season [15]. In this present study from eastern Taiwan, winter-linked occurrence of AMI showed an increased tendency but did not reach a significant level in the older group which accounted for 90% of all cases. However, the young group aged <45 years did have a greater occurrence of AMI during winter as compared to the other seasons. Possible explanations for the discrepancy in the seasonal variation in the occurrences of AMI between these studies may be a result of different geographic scopes, sample sizes, and age groups.

Findings of previous studies regarding age interaction in association of seasonality with AMI was controversial. In cold and temperate climate zones, multiple studies suggested that the older age (≥65 years) had a stronger association with AMI in the cold season when compared to the younger age (<65 years) [14,23]. In our previous study, the ≥65 years-linked stronger association with the cold season with AMI was in the subtropical region of western Taiwan but not in the tropical region [15]. These results provide evidence that suggests that the effects of advanced age tend to have greater influence on season-related incidences of AMI in the cooler regions. In contrast, a Korean study suggested that the younger age (<65 years) was more sensitive to seasonality in the occurrence of AMI compared to the older age [11]. In a similar way, the young adults aged <45 years showed a stronger association with winter-linked AMI compared to the older adults in this present study from eastern Taiwan.

Other than age, co-morbid risk factors may also interact in conjunction with season-related incidences of AMI. A study from Hong Kong revealed DM patients showed a greater frequency of admission for AMI compared to non-DM patients, markedly in the extreme temperature spectrum of both cold and hot seasons [14]. Our previous study suggested that metabolic dysfunction acted to modify the incidences of AMI between the cold and hot seasons, markedly in the tropical region of western Taiwan [15]. Smoking, obesity, DM, and hypertension are well-established risk factors for metabolic dysfunctions and cardiovascular diseases. A higher scale in the prevalence of smoking and BMI but lower in co-morbid DM and hypertension among the young adults diagnosed with AMI when compared to the older adults was documented consistently in previous and present studies [1,2,3]. For this reason, metabolic dysfunction in the young adults diagnosed with AMI appears be influenced more by smoking and obesity and less by DM and hypertension. Furthermore, triglycerides and total cholesterol might play an important role in metabolic dysfunction in the young adults diagnosed with AMI, as they were greater in the young group than the older group in this study. Based on the results mentioned above, the effect of metabolic dysfunction upon the occurrence of AMI in young adults is different from that in older adults and appears to be involved in young adult-linked increased risks of AMI during winter.

Evidence has indicated that smoking is an important and independent risk factor for AMI in young adults [18]. Through an analysis including over 230,000 participants in 15 countries across the Northern and Southern Hemisphere, Marti-Soler et al. found that cardiovascular risk factors and 10-year risk of dying from cardiovascular diseases showed a seasonal pattern characterized by relatively higher levels during winter [8]. Absolute risk difference of the 10-year risk of dying from cardiovascular diseases between the peak and nadir months among male smokers was 2.5 times greater than that among male non-smokers in the 40′s age group but just about 2 times in every age groups ≥50s [8]. In this present study, the rate of smoking was up to 71.7% in young adult men diagnosed with AMI and 51.1% in older adult men. A high proportion of smokers may be one possible explanation for the increased winter incidence of AMI in young adult men, as smokers have much elevated risks of severe cardiovascular diseases in winter relative to non-smokers, markedly among young adult men.

Along with smoking, obesity markedly associates with coronary heart disease in young adults [22,24]. Based on the guideline as defined by the World Health Organization in the Western Pacific, normal weight is a BMI (kg/m^2^) ranging 18.5–22.9, overweight is 23.0–29.9, and obesity is ≥30.0. A recent prospective study showed that obesity and weight gain was linearly associated with increased hazard ratio for coronary artery diseases among young adults [22]. In accordance to previous findings that a relatively higher BMI among the young AMI patients, BMI among the young group (26.8 ± 5.4 kg/m^2^) was greater than that among the older group (24.7 ± 4.8 kg/m^2^) in the present study. Of note is the finding by Marti-Soler et al. that BMI markedly increased from a nadir in summer to a peak in winter, ranging from 0.26 to 0.52 kg/m^2^ across 15 countries in the world [8]. A slight weight gain from summer to winter can become a trigger for AMI among obese young adults as they have a greater hazard ratio. Based on these findings, the excess risks of AMI in winter among young adult men was attributable to weight gain.

As winter brings a relatively lower temperature and shorter duration of sunlight, individuals would make not only physiological but also behavioral adjustments for the changes in environmental surroundings. In this present study conducted in a region with warm climate, winter-induced increases in AMI hospitalizations in young adult men are unlikely to be linked to a lower temperature, as the average temperature on the date of onset during winter was similar in the young group and the older group. The most likely explanation for increased incidence of AMI in young adult men in winter is behavioral changes in diet resulting in a higher intake of fat-rich foods and a reduction in physical activities. Previous findings indicated that winter weight gain was mainly body fat as a result of substantial increases in the intake of fat-rich foods such as meat and dairy products during winter as compared to summer [21,25]. Consequently, total serum cholesterol was elevated in winter, though total caloric intake did not show significant changes between winter and summer [24]. Evidence also showed that the winter environment resulted in reduced physical activities, especially for outdoor leisure activities. The impact of winter-related reduction in physical activities was marked among those of a younger age and the obese [10,26].

For AMI prevention among young adults, a non-smoking and smoke-free environment is recommended as first priority. Over the last few decades, the prevalence of smoking has decreased, while an increasing trend in overweight and obesity at a younger age has been observed across many countries and regions [24]. Multiple longitudinal studies demonstrate that obesity persists from childhood into adulthood [27]. Evidence supports that overweight and obesity from childhood onwards is associated with increased risks of coronary heart disease, while weight control in childhood and adolescence may reduce the risks [27,28]. Prevention of AMI among young adults should begin early in childhood through healthy diet and exercise leading to normal growth and weight gain. To avoid potential short-term winter triggers, young adults who are obese and/or smokers should be advised to take precautionary measures to control the intake of fat-rich foods and maintain adequate physical activities during winter months. Consistent participation in physical activity without seasonal interruption is a practical way to reduce risk of acute coronary events [29].

A strength of the current study is the low air pollution in the study region, as the effect of air pollution was not a confounding factor for AMI. Eastern Taiwan is a region free from industrial pollution, and the air quality in this region is good across the four seasons. Additionally, Taiwanese people are included in the compulsory health insurance system of medical care. The main limitation of this retrospective chart reviewing study is the possibility that medical records are incomplete or missing data. Thirty-eight percent of the overall sample (875 cases) are not included in this analysis due to lack of complete records that are required for the analysis. The results of comparison between 1412 cases that are included in the study and 875 cases that are not included show a significantly difference in age, the ratio of the aged <45 years to the aged ≥45 years, and BMI (see Appendix A). This study also suffers the limitation of small sample size in the young group, as AMI in young adults is uncommon. Small sample size may decrease statistical power. Waist circumference was not available in this present study, as some studies claim that waist circumference is more useful than BMI in analyzing obesity-related health risks; this weakness could not be addressed. This study further suffers from the inability to include analysis of females with AMI, as the sample amount is too small to allow for meaningful statistics. Due to the gender specificity of the study, it is not possible to generalize the findings to the population. Furthermore, the sample data were collected from one geographical region with a distinctly tropical climate. The warm climate of Taiwan does not allow for the generalization of these results to other climatic regions.

## 5. Conclusions

Male AMI patients under the age of 45 years are more likely than those over the age of 45 years to be smokers and obese. In contrast, AMI patients over 45 years are more likely than young AMI patients (<45 years) to be co-morbid for hypertension and/or DM. An effect due to seasonal changes on the occurrences of AMI is not apparent among adult men ≥45 years in a region without temperature extremes. However, occurrences of AMI are more prone to occur during winter among adult men aged <45 years. Temperature-related occurrences of AMI remain inconclusive; however, in combination with other complicating factors, increased risks for AMI during winter months are suggested. The identification of predictable and modifiable AMI triggers specifically for susceptible young adults is important in order to prevent the devastating impact on patients and their families. Further studies are required to extend understanding in different climatic regions and elucidate the possible negative effects of seasonal changes on the cardiovascular system.

## Figures and Tables

**Figure 1 ijerph-17-06140-f001:**
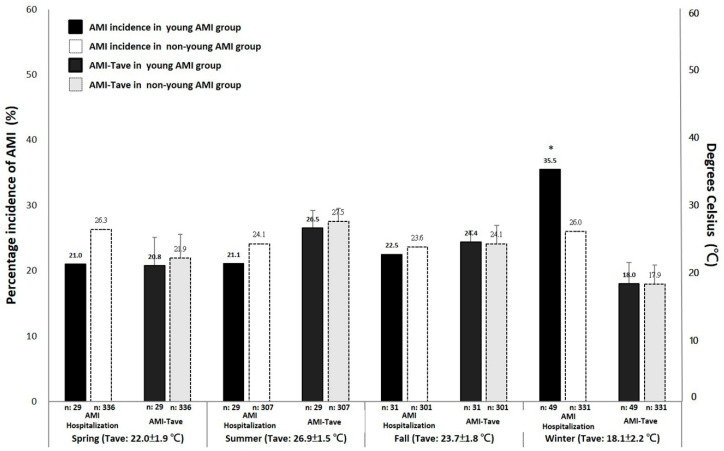
Seasonal distribution of AMI hospitalizations and the average temperature (Tave) on the date of AMI hospitalization (AMI-Tave) in the under-45 group and the over-45 group. * indicates *p* < 0.05 vs. spring, summer, and fall.

**Table 1 ijerph-17-06140-t001:** Comparison of clinical characteristics between the young group and the older group.

	The Young (*n* = 138)	The Older (*n* = 1275)	*p* Value
Age (SD)	39.1 (5.2)	66.7 (11.1)	<0.001 *
CAG (%)	128 (92.8)	1093 (85.7)	0.023 *
Number of stenosed vessels			0.001 *
≤1 (%)	77 (60.2)	489 (44.7)	
≥2 (%)	51 (39.8)	604 (55.3)	
Conventional risk factors			
Smoking (%)	99 (71.7)	651 (51.1)	<0.001 *
DM (%)	28 (20.3)	469 (36.8)	<0.001 *
Hypertension (%)	52 (37.7)	841 (66.0)	<0.001 *
T C (mg/dL)	183.0(62.0)	172.0(55.0)	0.002 *
HDL-C (mg/dL)	38.0(17.3)	39.0(14.0)	0.119
TG (mg/dL)	140.0(139.5)	106.0(91.0)	<0.001 *
LDL-C (mg/dL)	107.5(58.4)	106.8(48.0)	0.505
BMI (kg/m^2^)	26.8(5.4)	24.7(4.8)	<0.001 *

AMI: acute myocardial infarction; CAG: coronary angiography; DM: diabetes mellitus; TC: total cholesterol; TG: triglycerides; HDL-C: high-density lipoprotein cholesterol; LDL-C: low-density lipoprotein cholesterol; BMI: body mass index. * indicates *p* < 0.05.

**Table 2 ijerph-17-06140-t002:** Logistic regression model on the association of the season (winter/non-winter, spring/non-spring, summer/non-summer, and fall/non-fall, respectively) with the age group (the young/the older) after adjustment for the clinical characteristic factors which reached statistical difference between two age groups.

Multivariate Logistic Regression		
Main Predictor Variables	Adjusted Odd Ratio (OR) (95% CI)	*p* Value
Winter vs. Non-winter	1.750 (1.151–2.259)	0.009 *
Spring vs. Non-spring	0.727 (0.450–1.174)	0.129
Summer vs. Non-summer	0.685 (0.424–1.107)	0.123
Fall vs. Non-fall	1.043 (0.656–1.660)	0.857

* indicates *p* < 0.05.

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
