# Peer review of "Acute Myocardial Infarction among Young Adult Men in a Region with Warm Climate: Clinical Characteristics and Seasonal Distribution"

_ijerph, 2020, doi:10.3390/ijerph17176140_

Round 1

Reviewer 1 Report

Thank you for the opportunity to review the manuscript entitled " Acute Myocardial Infarction among Young Adult Men in a Region with Warm Climate: Clinical Characteristics and Seasonal Distribution". This is a very interesting and important manuscript.

In the present paper, authors investigated the influence of the seasons on AMI hospital admissions among the young (<45 years) and older (≥45 years) cohorts in eastern Taiwan.  Although the topic is of general interest, I have serious concerns:

  1. There are some unclear aspects in the methods. No extensive information is provided regarding the sampling procedures. What was the response rate?  Are all the people were taken from January 1994 to December 2015?  I think that the number of 1,413 admitted patients are not all the patients between so long follow up……  Please add the information. 
  2. This reviewer believes there were more analyses that could have been completed and reported. For instance:
  • You may consider using a time series analysis to compare AMI between two gtoups of age, or using a Poisson regression.
  • You need to fit a logistic regression model (overall with interaction between age and season and age groups stratification) and you must to adjust  to  the model to the risk factors such as smoking, DM, hypertension, total cholesterol, triglycerides, HDL-C, low-density lipoprotein cholesterol , body mass index.

  1. Discuss about the limitations (sample and selection bias, information bias, generality of this findings). The results may be difficult to generalize to other populations….

Abstract:

  1. Add the sentence -aim of the study.
  2. Lines 20-21: "The under-45 group showed significantly higher smokers, BMI, total cholesterol, and triglycerides but lower diabetes and hypertension than the under-45 group (p<.05)." The sentence is unclear.
  3. "Seasonal modulation of behaviors linked to smoking and obesity, rather than 27 temperatures, increases susceptibility to AMI during winter among young adult men at high-risk in 28 a region without temperature extremes."- How does this relate to the findings of the current study?.

Methods:

  1. Why you did not included the women?
  2. In the table 1: Please erase distribution of men, since all of them were men and the row with range (years). What is the minimum and maximum age of all age group?
  3. You wrote that "the data were expressed as mean ± SD for continuous variables and as proportions for categorized variables. Student's t test for continuous variables and Chi-square test for categorized variables were used to evaluate the differences between the two groups." But if the distribution of some variables strongly skewed, in which case, the median and the interquartile range need to be provided as well as appropriate nonparametric tests.
  4. You wrote that "A logistic regression model 94 was performed to assess the association of seasonal incidence with AMI for age stratification (<45y/o 95 / ≥45y/o)." but in table 2 was depicted only one logistic regression model on the association of the season with the under-45 group relative to the over-45 group. There is not stratification for age groups, may be do you mean seasonal stratification?

Conclusions:

  1. - How does this relate to the findings of the current study? -"The adverse effects of adaptive behaviors of eating fat-rich foods and reduced 256 physical activity during winter months, rather than the body’s reaction to low temperatures, are 257 likely risk factors linking to increased winter-induced incidence of AMI among adult men <45 years

Author Response

Major

Point 1: There are some unclear aspects in the methods. No extensive information is provided regarding the sampling procedures. What was the response rate?  Are all the people were taken from January 1994 to December 2015?  I think that the number of 1,413 admitted patients are not all the patients between so long follow up……  Please add the information. 

Response 1: Thank the reviewer for your comments and suggestions. This is a retrospective and cross-sectional study. Study cases was obtained from medical records. Hualien Tzu Chi Hospital is a medical center that accepts most of the transferred AMI patients in eastern Taiwan. From January 1994 to December 2015, data was accessed on 3378 individuals admitted to the hospital with a first-time diagnosis of AMI, including 2287 men (191 <45y/o and 2096 ≥45y/o) and 1091 women (27 <45y/o and 1064 ≥45y/o). The female population in the <45y/o age group was too small for meaningful statistical analysis over four seasons. As such, this study did not include female cases. Of the 2287 men, 875 had incomplete records, date of admission could not be accurately assessed and lacked one or more records of conventional risk factors. The 1413 with complete data were enrolled in this study and assigned to two groups: The young group (n=138, <45y/o) and the older group (n=1275, ≥45y/o). The information has been added in the section of 2.1. Study cases.

Point 2: This reviewer believes there were more analyses that could have been completed and reported. For instance:

  1. You may consider using a time series analysis to compare AMI between two gtoups of age, or using a Poisson regression.

Response 2-1: Thank the reviewer for the comments. As this is a cross-sectional study and all variables including age, Number of stenosed vessels, CAG, and Conventional risk factors were collected at the time of onset of AMI, the time series analysis is not suitable. For the suggested Poisson regression, the Poisson regression is used to analyze if the number of events is associated with specific condition (i.e., risk factor). Since the study purpose was to evaluate the association of season and age group (<45y/o vs. ≥45y/o) among AMI subjects, the logistic regression with dependent variable of age group and independent variable of season was used to analyze this relationship.

  1. You need to fit a logistic regression model (overall with interaction between age and season and age groups stratification) and you must to adjust to the model to the risk factors such as smoking, DM, hypertension, total cholesterol, triglycerides, HDL-C, low-density lipoprotein cholesterol, body mass index.

Response 2-2: Thank the reviewer for the comments. The logistic regression analysis has been conducted with the results summarized in Table 2. Because all subjects in the current study were AMI subjects and the purpose of the study was to evaluate the association of season and age group (the young vs. the older) among AMI subjects, four models were built with dependent variable of age group and independent variable of each season including winter vs. non-winter, spring vs. non-spring, summer vs. non-summer, fall vs. non-fall, respectively. In the models, the risk factors such as smoking, DM, hypertension, total cholesterol, triglycerides, HDL-C, low-density lipoprotein cholesterol, body mass index are included in the model for adjustment according to your suggestion.

  1. Discuss about the limitations (sample and selection bias, information bias, generality of this findings). The results may be difficult to generalize to other populations….

Response 2-3: Thank the reviewer for this helpful suggestion. The suggestion has been updated. Please refer to the paragraph regarding to the limitation for the reversion (Lines 257-262).

Abstract

Point 1: Add the sentence -aim of the study.

Response 1: Thank the reviewer for the suggestion. The abstract has been edited to include your suggestion in the first sentence.

Point 2: Lines 20-21: "The under-45 group showed significantly higher smokers, BMI, total cholesterol, and triglycerides but lower diabetes and hypertension than the under-45 group (p<.05)." The sentence is unclear.

Response 2: Thank the reviewer for the reminding. The sentence has been edited for clarity as" The young group showed significantly higher percentage of smokers, BMI, total cholesterol levels, and triglycerides levels but lower percentage of diabetes and hypertension than the older group (p<.05)." Please refer to the section of abstract for the reversion (Lines 21-23).

Point 3: "Seasonal modulation of behaviors linked to smoking and obesity, rather than 27 temperatures, increases susceptibility to AMI during winter among young adult men at high-risk in 28 a region without temperature extremes."- How does this relate to the findings of the current study?

Response 3: Thank the reviewer for this inquiry. We have deleted this sentence, as the previous sentence has provided a brief conclusion regarding to the findings of this study.

Methods

Point 1: Why you did not included the women?

Response 1: Thank the reviewer to ask and for allowing us to explain more regarding this issues. The reason why the studied population emphasizes on men is based on the findings that AMI in young adults is uncommon and mainly among men, with an increasing incidence in recent years [ref. 16,17]. This information has been added in the section of introduction (Line 57-58). This study further suffers from the inability to include analysis of females with AMI as the sample amount is too small to allow for meaningful statistics. This explanation has been described in the section of 2.1. Study cases (Line 68-72) and the paragraph regarding to limitation (Line 257-262).

Point 2:  In the table 1: Please erase distribution of men, since all of them were men and the row with range (years). What is the minimum and maximum age of all age group?

Response 2: Thank the reviewer for this suggestion. The rows of gender and range (years) in the table 1 have been deleted, and the minimum and maximum age of all age group have been added in the row of age.

Point 3:  You wrote that "the data were expressed as mean ± SD for continuous variables and as proportions for categorized variables. Student's t test for continuous variables and Chi-square test for categorized variables were used to evaluate the differences between the two groups." But if the distribution of some variables strongly skewed, in which case, the median and the interquartile range need to be provided as well as appropriate nonparametric tests.

Response 3: Thank the reviewer for the comments. The Kolmogorov-Smirnov test has been conducted to evaluate the skewness of the continuous variable according to your suggestion. The results showed that the distributions of all continuous variables in Table 1 were skewed, we performed the Mann-Whitney test to test the difference of these variables between age groups instead of Student's t test. The median and the interquartile range have also been provided. Please refer to Table 1 and section 2.4 for the revision.

Point 4:  You wrote that "A logistic regression model 94 was performed to assess the association of seasonal incidence with AMI for age stratification (<45y/o 95 / ≥45y/o)." but in table 2 was depicted only one logistic regression model on the association of the season with the under-45 group relative to the over-45 group. There is not stratification for age groups, may be do you mean seasonal stratification?

Response 4: Thank the reviewer for the comments. What we meant was that a logistic regression model was performed to assess the association of seasonal and age group (<45y/o / ≥45y/o) among the AMI subjects. Please refer to section 2.4 for the correction.

Conclusions

Point 1: How does this relate to the findings of the current study? -"The adverse effects of adaptive behaviors of eating fat-rich foods and reduced 256 physical activity during winter months, rather than the body’s reaction to low temperatures, are 257 likely risk factors linking to increased winter-induced incidence of AMI among adult men <45 years

Response 1: Thank the reviewer for the comments. In order to avoid the confusion, the conclusion regarding this sentence has been revised as “Temperature related occurrences of AMI remain inconclusive, however in combination with other complicating factors results in increased risk for AMI during winter months.” Please refer to the section of conclusion for the revision (Line 269-271).

Reviewer 2 Report

Shih CY et al. conducted research work to determine the associations between seasons and AMI incidence by age (< 45 years vs. ≥45 years) among 1,413 hospitalized men from 1994 to 2015 with AMI clinical diagnosis at the first time. This is a cross-sectional study based on conventional risk factors from medical records and weather data from the Taiwan Central Weather Bureau. The authors first showed clinical characteristics. Secondly, the authors determined the mean ± SD of AMI incidence rate and temperature at the date of hospitalization (AMI-Tave) by age 45 across four seasons. Last, the author tested the associations of seasons with AMI incidence between age <45 and ≥45 groups by logistic regression analysis. Overall, this article is in good shape with improvement needed. Below are my suggestions and questions.

1) Please use the consistent term for the groups, e.g., young vs. older adults or young vs. non-young adults.

2) The author kept using “cohort” in the article. Is this a cohort study? Did the author follow patients over time? Please clarify the study type.

3) How did the authors calculate AMI incidence rate among AMI patients for Figure 1?  

4) Too many assumptions in the conclusion part. To describe the conclusion like that, the authors need to provide data regarding associations of AMI incidence with smoking, obesity, and adaptive behaviors in winter among patients less than age 45.

Author Response

Point 1: Please use the consistent term for the groups, e.g., young vs. older adults or young vs. non-young adults.

Response 1: Thank the reviewer for this suggestion. " young vs. older adults" is an excellent term for the groups. " young vs. older adults" has been edited for the groups throughout this manuscript.

Point 2: The author kept using “cohort” in the article. Is this a cohort study? Did the author follow patients over time? Please clarify the study type.

Response 2: We apologize for the confusion we made. This is not a cohort study. "cohort" is not a proper word in this article. "cohort" has deleted or replaced by "group" throughout this manuscript.

Point 3: How did the authors calculate AMI incidence rate among AMI patients for Figure 1?

Response 3: We apologize for the confusion we made. “AMI hospitalization” is more proper term than “AMI incidence” in this article. "hospitalization" has been used to revise “incidence” in Figure 1.

Point 4:  Too many assumptions in the conclusion part. To describe the conclusion like that, the authors need to provide data regarding associations of AMI incidence with smoking, obesity, and adaptive behaviors in winter among patients less than age 45.

Response 4: Thank the reviewer for this comments. In order to avoid the confusion, the conclusion regarding these assumptions has been revised. Please refer to the section of conclusion for the revision.

Round 2

Reviewer 1 Report

Reviewer believes that the manuscript has improved. However, some attention to fine detail and sentence structure is warranted. Below are some suggestions.

1. Abstract: The aim of this study was to investigate the influence of the seasons on acute myocardial 12 infarction (AMI) among young adult men. I advise to add : " The aim of this cross sectional study was to investigate the influence of the seasons on acute myocardial 12 infarction (AMI) among young adults aged <45 years compared to old adults aged ≥45 years "

2. Abstract: Data were extracted from 1413 male AMI 14 patients from January 1994 to December 2015, including onset date, average temperature at onset (AMI-Tave),… " Please add what is Tave to the abstract.

3. Abstract: I think that it is not correct: "Logistic regression analyses were used to evaluate the associations between the seasons and the 20 young group." The right sentence is :" Logistic regression analyses were used to evaluate the associations between the seasons and the AMI hospitalization among "

4. Introduction: the sentence is not clear " Previous studies have consistently reported that AMI in young adults is uncommon and mainly 57 among men, with an increasing incidence in recent years"

5. Introduction: "The AMI hospitalization rate and average temperature at date of onset were assessed according to 61 the seasons." Move the sentence to the methods.

6. Methods: "Of the 2287 men, 875 had incomplete records, date of admission could not be accurately assessed and lacked one or more records of conventional risk factors."

38% of the overall sample had incomplete records.  this is a significant limitation which should be mentioned in discussion and I think that is very important to add supl. Table and compare variable distribution between 1412 men who included in the study to men who did not included.

7. Methods: "A logistical regression model was performed to assess the association between season and age group (the young / the older) with their conventional risk factors." THE SENTENCE IS NOT CLEAR.

8. Table 2: Logistic regression model on the association of the season with the young group relative to the older group after adjustment for conventional risk factors. Please add which for which factors. How the predictors were chosen in the final multivariable logistic regression.

My understanding from the methods that you did 4 logistic regressions for each season. However I don’t understand how you did the comparison of young group relative to  the older group? I think that two multivariable logistic regression models should be fitted separately for each age group in which the season variable is one dummy variable.

Author Response

Response to Reviewer 1 Comments

Major

Point 1: Abstract: The aim of this study was to investigate the influence of the seasons on acute myocardial 12 infarction (AMI) among young adult men. I advise to add : " The aim of this cross sectional study was to investigate the influence of the seasons on acute myocardial 12 infarction (AMI) among young adults aged <45 years compared to old adults aged ≥45 years " 

Response 1: Thank the reviewer for the advice. The suggested information has been added in the first sentence of the abstract.

Point 2: Abstract: Data were extracted from 1413 male AMI 14 patients from January 1994 to December 2015, including onset date, average temperature at onset (AMI-Tave),… " Please add what is Tave to the abstract.

Response 2: Thank the reviewer for the suggestion. “average temperature at onset (AMI-Tave)” has been rewritten as “the average temperature (Tave) on the date of AMI hospitalization (AMI-Tave)”. Please refer to Line 16-17 for the revision.

Point 3: Abstract: I think that it is not correct: "Logistic regression analyses were used to evaluate the associations between the seasons and the 20 young group." The right sentence is :" Logistic regression analyses were used to evaluate the associations between the seasons and the AMI hospitalization among "

Response 3: Thank the reviewer for the suggestion. The sentence has been rewritten as the suggested. Please refer to Line 22 for the revision.

Point 4: Introduction: the sentence is not clear " Previous studies have consistently reported that AMI in young adults is uncommon and mainly 57 among men, with an increasing incidence in recent years"

Response 4: Thank the reviewer for your comment. The sentence has been rewritten and edited to the end of first paragraph in the section of introduction. Please refer to Line 40-43 for the revision.

Point 5:  Introduction: "The AMI hospitalization rate and average temperature at date of onset were assessed according to 61 the seasons." Move the sentence to the methods.

Response 5: Thank the reviewer for the suggestion. The sentence has been moved to the end of the section of 2,3. Please refer to Line 100-101 for the revision.

Point 6:  Methods: "Of the 2287 men, 875 had incomplete records, date of admission could not be accurately assessed and lacked one or more records of conventional risk factors." 38% of the overall sample had incomplete records. this is a significant limitation which should be mentioned in discussion and I think that is very important to add supl. Table and compare variable distribution between 1412 men who included in the study to men who did not included.

Response 6: Thank the reviewer for the comment. Comparison of variable distribution between 1412 cases who were included in the study and 875 cases who were not included was shown in Table S1. This has been added in the section of 2.1 Study cases (Line 77-78). The main limitation of this retrospective chart reviewing study is the possibility that medical records are incomplete or missing data. 38% of the overall sample (875 cases) are not included in this analysis due to lack of complete records that are required for the analysis. The results of comparison between 1412 cases who were included in the study and 875 cases who were not included show a significant difference in age, the ratio of the aged <45 years to the aged ≥45 years, and BMI. The limitation mentioned above has been addressed in the section of discussion (Line 269-274).

Point 7:  Methods: "A logistical regression model was performed to assess the association between season and age group (the young / the older) with their conventional risk factors." THE SENTENCE IS NOT CLEAR.

Response 7: Thank you very much for your comments. The logistic regression analysis was performed to analyze the season effect on the age (the young / the older) at onset of AMI. In the analysis, age group (the young/ the older) was set as the dependent variable. Four logistic regression models were then built by using Winter/Non-winter, Spring/Non-Spring, Summer/Non-summer, and Fall/Non-fall as the main predictor, respectively. We have described the corresponding description clearly. Please refer to the section of 2.4 Statistical analysis (Line 108-115) for the revision.

Point 8: Table 2: Logistic regression model on the association of the season with the young group relative to the older group after adjustment for conventional risk factors. Please add which for which factors. How the predictors were chosen in the final multivariable logistic regression. My understanding from the methods that you did 4 logistic regressions for each season. However, I don’t understand how you did the comparison of young group relative to the older group? I think that two multivariable logistic regression models should be fitted separately for each age group in which the season variable is one dummy variable.

Response 8: Thank you very much for your comments. The main purpose of study was to assess the season effect on the age (the young / the older) of study cases at AMI hospital admissions. Therefore, age group (the young/ the older) was set as the dependent variable in the logistic regression models. Four logistic regression models were then built by including Winter/Non-winter, Spring/Non-Spring, Summer/Non-summer, and Fall/Non-fall as the main predictor, respectively. We have rewritten the descriptions of the related parts. Please refer to the section of 2.4 (Line 108-115), the section of 3.3 (Line 142-149), and Table 2 for the revision.

Reviewer 2 Report

This manuscript has improved significantly. I have a few more suggestions and questions, as shown below.

  1. Please show Age (SD) or Age (range) in Table 1.
  2. Please show statistical results for Age and CAG between the young and older groups in Table 1.
  3. Please identify the reference groups in the footnote of Table 2. Based on the table, I assumed the author performed logistic regression analysis by comparing season and non-season. However, the title of Table 2 described the model tested the association of the young group relative to the older group and the article described the model tested association between season and age. Please edit the descriptions in the manuscript according to the results and the analytical approach of Table 2. 
  4. Is the number of stenosed vessels an effect modifier of the association between winter and AMI-hospitalization among the young group?

Author Response

Response to Reviewer 2 Comments

Point 1: Please show Age (SD) or Age (range) in Table 1.

Response 1: Thank the reviewer for the suggestion. Age(SD) has been shown in Table 1.

Point 2: Please show statistical results for Age and CAG between the young and older groups in Table 1.

Response 2: Thank the reviewer for the suggestion. P values of statistical results for Age and CAG between the young and older groups has been shown in Table 1.

Point 3: Please identify the reference groups in the footnote of Table 2. Based on the table, I assumed the author performed logistic regression analysis by comparing season and non-season. However, the title of Table 2 described the model tested the association of the young group relative to the older group and the article described the model tested association between season and age. Please edit the descriptions in the manuscript according to the results and the analytical approach of Table 2. 

Response 3: Thank you very much for your comments. The main purpose of study was to assess the season effect on the age (the young / the older) of study cases at AMI hospital admissions. In order to avoid the confusion, the title of Table 2 has been revised as “Logistic regression model on the association of the season (Winter/Non-winter, Spring/Non-Spring, Summer/Non-summer, and Fall/Non-fall respectively) with the age group (the young/the older) after adjustment for the clinical characteristic factors which reached statistical difference between two age groups.” We also have rewritten the descriptions of the related parts in the section of 2.4 (Line 108-115) and the section of 3.3 (Line 142-149).

Point 4: Is the number of stenosed vessels an effect modifier of the association between winter and AMI-hospitalization among the young group?

Response 4: Thank the reviewer for the comment. Our answer is yes! The number of stenosed vessels assessed by coronary angiography indicates the extent of atherosclerotic change within coronary arteries. The occurrences of AMI in different age groups and seasons may be influenced by the severity of atherosclerotic condition. As reaching statistical difference between two age groups, the number of stenosed vessels is included as one of adjustment variables for the multivariate logistic regression analysis that assesses the association of the season (winter/Non-winter) with the age group (the young/the older). Please refer to the section of 2.4 (Line 113-114) and the section of 3.3 (Line 145-149).
